# Identifying recharge under subtle ephemeral features in flat-lying semi-arid region using a combined geophysical approach

Brady A. Flinchum[1], Eddie Banks[2], Michael Hatch[2,3], Okke Batelaan[2], Luk Peeters[1], Sylvain Pasquet[4]

[1]Commonwealth Scientific Industrial Research Organization (CSIRO), Deep Earth Imaging Future Science Platform & Land and Water, Urrbrae, 5064, Australia

[2] National Centre for Groundwater Research and Training, College of Science and Engineering, Flinders University, Adelaide, 5001, Australia

[3] Department of Geosciences, School of Physics, University of Adelaide, Adelaide, Australia

[4] Université de Paris, Institut de physique du globe de Paris, CNRS, F-75005 Paris, France.

*Correspondence to*: Brady A. Flinchum (brady.flinchum@csiro.au)

**Abstract.** Identifying and quantifying recharge processes linked to ephemeral surface water features is challenging due to their episodic nature. We use a combination of well-established near-surface geophysical methods to provide evidence of a surface and groundwater connection under a small ephemeral recharge feature in a flat, semi-arid region near Adelaide, Australia. We use a seismic survey to obtain P-wave velocity through travel-time tomography and S-wave velocity through the multichannel analysis of surface waves. The ratios between P-wave and S-wave velocities are used to calculate Poisson's ratio, which allow us to infer the position of the water table. Separate geophysical surveys were used to obtain electrical conductivity measurements from time-domain electromagnetics and water contents from downhole nuclear magnetic resonance. The geophysical observations provide evidence to support a groundwater mound underneath a subtle ephemeral surface water feature. Our results suggest that recharge is localized and that small-scale ephemeral features may play an important role in groundwater recharge. Furthermore, we show that a combined geophysical approach can provide a perspective that helps shape the hydrogeological conceptualization of a semi-arid region.

## 1 Introduction

Understanding groundwater recharge mechanisms and surface water-groundwater connectivity is crucial for sustainable groundwater management (Banks et al., 2011; Brunner et al., 2009). In semi-arid areas, recharge has been shown to occur in focused regions beneath perennial streams and lakes, and ephemeral streams and ponds (Cuthbert et al., 2016; Scanlon et al., 2002, 2006). However, identifying localized regions of groundwater recharge remains challenging.

Many aquifers in semi-arid areas receive a significant portion of their recharge from adjacent mountain ranges (Bresciani et al., 2018; Earman et al., 2006; Wilson and Guan, 2004; Winograd et al., 1998). In this common scenario, recharge can occur via groundwater flow from the mountain range directly into the aquifer—implying a significant lateral groundwater connection with the adjacent mountain range (Markovich et al., 2019). Alternatively, precipitation from the mountain range

flows out and across the semi-arid basin as surface water and recharge the aquifer via river infiltration processes—implying a vertical connection between surface and groundwater (Bresciani et al., 2018; Brunner et al., 2009; Winter et al., 1998).

Groundwater recharge processes span a wide range of spatial and temporal scales making them difficult to quantify (Scanlon et al., 2002). Recharge rates are traditionally quantified using physical, tracer, or modelling techniques (Scanlon et

al., 2002). Physical techniques include carefully measuring fluxes and evapotranspiration along various reaches of a river or stream (Abdulrazzak, 1995; Lamontagne et al., 2014), by calculating aquifer water level response times (e.g. water table fluctuation method) (Cuthbert et al., 2019), or  through stream hydrograph separation (Banks et al., 2009; Chapman, 1999; Cuthbert et al., 2016). Common tracer techniques include the use of stable isotopes of hydrogen and oxygen (Lamontagne et al., 2005; Taylor et al., 1992; Winograd et al., 1998), quantifying chemical signatures that have accumulated from past human

activities (e.g. chlorofluorocarbons and sulphur hexafluoride) (Cook et al., 1996), and measuring environmental tracers such as chloride (Allison et al., 1990; Anderson et al., 2019; Crosbie et al., 2018) and radon (Bertin and Bourg, 1994; Genereux and Hemond, 2010; Hoehn and Gunten, 1989). Lastly, numerical modelling is used to estimate recharge over global scales (Gleeson et al., 2012; Scanlon et al., 2006) and test existing hydrogeological conceptualizations (Xie et al., 2014).

Quantifying recharge processes in ephemeral ponds or streams in semi-arid regions is particularly difficult because

flooding events are episodic (Shanafield and Cook, 2014). The infrequency and variable size of flooding events makes it difficult to monitor, quantify, or even identify if groundwater recharge has occurred. Furthermore, infiltration is a different process than recharge. Groundwater recharge must be confirmed by a response in the water table, whereas water that has infiltrated might have been taken up by vegetation or lost to evaporation. Larger ephemeral rivers flood frequently so equipment can be installed and be ready when an event occurs (Dahan et al., 2007, 2008). On the other hand, it is more difficult

to capture recharge events of smaller ephemeral tributaries; as a result, the recharge mechanisms of these features are less understood. These smaller scale features are common on Earth's surface. It has been shown that 69% of first-order streams and ~34% of larger fifth-order rivers below 60° latitude are ephemeral (Acuña et al., 2014; Raymond et al., 2013). Thus, even if small ephemeral features only provide small amounts of groundwater recharge during individual events, their large spatial distribution means that they could be important to recharge processes of a given region.

Small ephemeral features are an ideal target for near-surface geophysical surveys. A wide range of existing and standardized geophysical techniques have been used in hydrological studies (e.g. Robinson et al., 2008; Siemon et al., 2009; Parsekian et al., 2015). To highlight surface and groundwater connections, geophysical methodologies commonly rely on time-lapse measurements. This is because the infiltration of groundwater causes changes in geophysical properties on the order of days or months (i.e. the geology stays constant). Time-lapse electrical resistivity measurements have been used to observe and

monitor recharge pathways (Carey and Paige, 2016; Singha and Gorelick, 2005; Johnson et al., 2012; Valois et al., 2016; Thayer et al., 2018; Kotikian et al., 2019) and can highlight preferential flow paths. These methods are still handicapped by the fact that they still require the burial or setup of the geophysical equipment prior to a natural recharge (Kotikian et al., 2019; Thayer et al., 2018) or a man-made event (Carey and Paige, 2016; Claes et al., 2019). It is still challenging to find a suitable

geophysical approach that can be deployed rapidly (that is without a time-lapse setup) to determine if an ephemeral drainage

feature is acting as a groundwater recharge feature.

The aim of this study is to use a combination of well-established near-surface geophysical methods to provide evidence of a surface and groundwater connection of a small, shallow, and subtle ephemeral feature in a low-lying semi-arid landscape without time-lapse measurements. We used a single seismic survey to obtain P-wave velocity through seismic refraction tomography (SRT) (Sheehan et al., 2005; Zelt et al., 2013) and S-wave velocity through the multi-channel analysis

of surface waves (MASW) (Park et al., 1999; Pasquet and Bodet, 2017) to calculate Poisson's ratio, which allowed us to infer the position of the water table. A separate survey was used to obtain bulk electrical conductivity measurements from time-domain electromagnetics (TEM) (Parasnis, 1986; Reynolds, 2011; Telford and Telford, 1976). Water contents and $T_2$ relaxation times (time constant for the decay of transverse magnetization) were acquired using downhole nuclear magnetic resonance (NMR) (Walsh et al., 2013). We used this combination of standard geophysical measurements to show that small-

scale ephemeral features are likely to contribute to the localized replenishment of groundwater in shallow unconfined aquifers in this low-lying semi-arid environment.

## 2 Site Description

The North Adelaide Plains (NAP) is located north of the city of Adelaide, Australia and is part of the St Vincent Basin, a geological basin underlying the area between the Yorke Peninsula and the Mount Lofty Ranges in South Australia

(Figure 1). The St Vincent Basin is a north-south trending basin that is characterized by low topographic relief between 0 and 200 m elevation above sea level (Smith et al., 2015). The NAP is bound by the Mount Lofty Ranges to the East and its northern boundary is marked by the Light River (Figure 1). Land-use in the NAP is predominantly dryland agriculture with mixed farming (sheep and rotational cropping of wheat, barley, and canola) (The Goyder Institute for Water Research, 2016). Potential evaporation is high and the average rainfall is low, averaging around 445 mm·$yr_{-1}$, with an average daily temperature

of 21.6 °C (Bresciani et al., 2018). The combination of low rainfall and high evaporation rates in the NAP implies that the source water in the aquifers is from the Mount Lofty Ranges where the average rainfall is 983 mm·$yr_{-1}$ (Bresciani et al., 2018). Rainfall is winter-dominated (May to August), which suggests that recharge is also seasonal (Batlle-Aguilar et al., 2017; Bresciani et al., 2018).

LiDAR of the NAP shows that within this low relief landscape there are many small ephemeral surface drainage

features (Figure 1). These subtle drainage features are visible in the hill-shaded LiDAR and indicates that surface water runoff is likely to flow towards these ephemeral drainage features and toward the larger streams after precipitation events (Figure 1b). These ephemeral features are not monitored because they fall below the resolution of the 30 m SRTM elevation data (Figure 1b). The near-surface Quaternary aquifers are typically used for stock and domestic purposes and have salinity ranges between 2000 and 13,000 mg·$L_{-1}$ (Department for Water, 2010; The Goyder Institute for Water Research, 2016). The near-

surface aquifers are only monitored because they present a risk of waterlogging and soil salinization (Department for Water, 2010).

Most of the water that recharges the NAP aquifers comes from the Mount Lofty Ranges to the East, supported by the fact that in between streams you find high-salinity water which suggests the occurrence of diffuse recharge (see conceptual model in Bresciani et al 2018 (Figure 14)). It has been shown on basis of multiple lines of evidence that water flows from the Mount Lofty Ranges onto the NAP through ephemeral rivers and streams and recharges the underlying aquifers via vertical infiltration (Bresciani et al., 2018). In this conceptual model, recharge is mostly localized and occurs along the main rivers and streams. This conceptual model is supported by lower groundwater chloride concentrations surrounding Gawler River and Little Para River in both the Quaternary and Tertiary aquifers and the piezometric surfaces that show groundwater moving away from the rivers (losing river conditions) and into the underlying aquifers (Bresciani et al., 2018). Prior to this study, it was argued that the aquifers of the NAP were recharged through a lateral groundwater connection with the rocks underlying the Mount Lofty Ranges. This concept was supported by an increase in groundwater ages away from the Mount Lofty Ranges and stable isotopes indicating some evaporation prior to infiltration (Batlle-Aguilar et al., 2017). It is important to note that the age and isotope data supports equally well the mountain front recharge conceptual model of Bresciani et al. (2018).

Our study site is located on a private farm, 44 km northwest of Adelaide and is between the Light and Gawler rivers (Figure 1). In May of 2018, as part of a regional study of the shallow water table, 47 shallow holes were drilled across the northern region of the NAP with a small truck-mounted Rockmaster drill rig (Hatch et al., 2019). We used these pre-existing sites to select a location where we knew the water table was within a range of 3-10 m to increase the likelihood of imaging the water table with the seismic data. The existing drillhole would also provide ground truthing to the geophysical data. Thus, our study transect for the near surface geophysical surveys was located adjacent to one of these drillhole sites where we had manual water level measurements, soil samples, and downhole NMR logs. The 235-m-long transect line was positioned so that it crossed a small ephemeral topographic feature that is only visible on a map with via high resolution elevation data collected via drone (Figure 1c).

## 3 Methods

To aid in geophysical interpretation and reduce ambiguities, it is important to "ground-truth" near-surface geophysical data with drilling results (Flinchum et al., 2018; Gottschalk et al., 2017; Orlando et al., 2016; West et al., 2019) or to corroborate them by other independent geophysical measurements. In this study, we combined hydrogeological observations with multiple geophysical measurements to obtain different geophysical parameters, specifically: bulk electrical conductivity from TEM, P-wave velocity from SRT, S-wave velocity from MASW, and water contents from downhole NMR. In April 2018, the shallow drillhole was logged with a downhole NMR system (Vista Clara Dart) and the water level was measured by hand. Only a week after the seismic data were collected, a separate campaign was carried out to collect 26 TEM soundings along the same profile (Figure 1c). In the following manuscript, we use these geophysical methods to infer a surface water-groundwater connection

without time-lapse measurements. In this section we briefly describe the theory behind the geophysical methods and how the measurements are influenced by various hydrological properties. Additional figures and details pertaining to the processing of the geophysical data set can be found in the supplementary material.

## 3.1 Topography Acquisition

At our study site, no LiDAR imagery was available. High resolution imagery of the small study area (~9 hectares) was thus acquired with a DJI Phantom 4 Pro unmanned aerial vehicle (UAV). The UAV flew a grid pattern over the study area at an elevation of 30 m above ground level and collected a photo dataset of 834 images. Georeferencing was undertaken using a Trimble R10 global positioning system (GPS) Real Time Kinematic (RTK) survey with 65 ground control points located within the study area and provided a georeferencing root mean square error (RMS) of 0.153 meters. The captured photos were processed using the photogrammetry Pix4D software package (Pix4Dmapper Pro version 3.2, 2017) to generate a high resolution (0.8 cm/pixel) digital surface model (DSM). As the study area was a fallow field at the time of the survey, the DSM was treated as a digital elevation model (DEM) as there was very little vegetation present. The generated DEM was re-sampled to a 0.5 m DEM (Figure 1) that was used to extract the elevation profile along the geophysical transect.

## 3.2 Seismic Refraction Tomography

Seismic refraction is an active source geophysical method that estimates seismic velocity. A seismic refraction survey provides a spatial distribution of P-wave velocity (energy propagating along the direction of travel). In a shallow seismic refraction survey, the time taken for the energy to travel from a source to each individual receiver, called a travel-time, is measured. The subsurface velocity structure controls the travel-times so they can be inverted to retrieve the subsurface P-wave velocity structure using a forward model and an inversion scheme (Sheehan et al., 2005; Zelt et al., 2013). P-wave velocity is controlled by the elastic properties of the material, porosity, and saturation (Berryman et al., 2002; Hashin and Shtrikman, 1963). If the pore space is filled with a fluid, in our case water (regardless of salinity), then the P-wave velocity is greater than if the pore space is not filled with fluid (Bachrach and Nur, 1998; Desper et al., 2015; Gregory, 1976; Nur and Simmons, 1969).

In this survey, we used 48 geophones spaced at 5 m, which produced a 235 m long profile. The source was a 40 kg accelerated weight, striking a 20 x 20 x 2 cm steel plate at every geophone. To increase the signal-to-noise, 8 shots were stacked at each of the 80 locations. The travel-times were picked manually (Figure S1 and S2) and inverted for P-wave velocity using the refraction module in the Python Geophysical Inversion and Modeling Library (pyGIMLi) (Rücker et al., 2017). The forward model is based on the shortest path algorithm (Dijkstra, 1959; Moser, 1991; Moser et al., 1992). PyGIMLi utilizes a deterministic Gauss-Newton inversion scheme and incorporates a data weight matrix (Rücker et al., 2017). We populated the data weight matrix using reciprocal travel-times (Figure S2). To initialize the inversion, we used a gradient model that had a velocity of 0.4 km·s-1 at the surface and 2 km·s-1 at a depth of 40 m. To quantify uncertainty, we incorporated a bootstrapping

algorithm on the travel-time picks (details in the supplementary material). The model fits are determined by a $\chi^2$ misfit, which incorporates our picking errors and a root mean square (RMS) error (details in supplementary material).

### 3.3 Multichannel Analysis of Surface Waves

At the Earth's surface, most of the elastic energy travels as surface waves. Surface waves are the largest amplitude events that are recorded in both active source seismic acquisition and earthquake records. Surface waves are caused by interactions of the body waves (P-waves and S-waves) and the boundary conditions that only exist at the surface (Stein and Wysession, 1991). There are two types of surface waves: Love waves and Rayleigh waves (for a detailed review on surface waves, the reader is referred to Stein and Wysession, 1991; Lowrie, 2007). In this study we take advantage of the dispersive nature of Rayleigh waves (Park et al., 1999; Pasquet and Bodet, 2017; Xia et al., 1999, 2003). Furthermore, Rayleigh waves propagate at velocities mostly driven by the S-Wave velocity of the medium. The dispersion of Rayleigh waves can be measured by picking the phase velocity as a function of frequency (Park et al., 1999; Xia et al., 2003). The phase velocity of lower frequencies (longer wavelengths) will be influenced by deeper S-wave velocity structures whereas higher frequencies will be influenced by shallower structures. These frequency dependent phase velocities can then be inverted for one-dimensional (1D) S-wave velocity models at low computational costs (Pasquet and Bodet, 2017).

In this study, we use the acquisition set up from the refraction survey to analyse the dispersion of surface wave energy. This approach produces a pseudo two-dimensional (2D) section comprised of 41 1D S-wave velocity profiles, spaced every 5 m starting at 17.5 m from the start of the profile. To build the pseudo 2D profile we used the Surface Wave Inversion and Profiling (SWIP) package (Pasquet and Bodet, 2017). First, the seismic data is resorted and windowed to sample 1D vertical slices of the subsurface. Once windowed, the sorted seismic data are transformed into the frequency-phase velocity domain using a slant stack (Mokhtar et al., 1988). To increase the depth of investigation, similar dispersion curves from different shots are stacked together (Neducza, 2007). Once the dispersion curves are constructed they are picked and an uncertainty associated with each pick is defined (O'Neil, 2003) (Figure S4). To construct our dispersion curves, we used 40 m windows (8 stations) and ensured a 5 m offset between the source and first channel to avoid near-source effects. The picks and corresponding uncertainty for each windowed dispersion curve are inverted using a Monte Carlo approach and the neighbourhood algorithm (Sambridge, 1999; Wathelet et al., 2004). We ran 15,000 inversions for each of our dispersion curves and averaged the 1000 best-fitting S-wave velocity models to build final 1D models (Figure S5) every 5 m (more details about processing can be found in the supplementary material). Finally, the individual 1D S-wave profiles are combined into a pseudo-2D section (Pasquet et al., 2015b, 2015a; Pasquet and Bodet, 2017).

### 3.4 Poisson's Ratio

Locating the water table of the unconfined aquifer over large spatial scales is challenging and is traditionally done by drilling down to the water table and interpolating manual water level measurements between drillhole locations. Building a detailed water table map requires many measurements and can be limited by logistical or financial constraints. Here, we exploit

190 the fact that P-wave velocities increase when a material is saturated and the S-wave velocities remain relatively unchanged (Bachrach and Nur, 1998; Desper et al., 2015; Gregory, 1976; Nur and Simmons, 1969).

  Poisson's ratio is a unitless elastic property that describes how much a material will deform in the direction perpendicular to an applied stress. Poisson's ratio can be calculated from P-wave and S-wave velocities (Eq. 1).

$$\upsilon = \frac{V_p^2 - 2V_s^2}{2(V_p^2 - V_s^2)}, \tag{1}$$

195 In Eq. 1, $V_p$ is P-wave velocity, $V_s$ is S-wave velocity and $\upsilon$ is Poisson's ratio. Poisson's ratio for geologic materials ranges from 0 to 0.5. Poisson's ratio increases as fluid saturation increases (Bachrach et al., 2000; Dvorkin and Nur, 1996; Nur and Simmons, 1969; Salem, 2000). Furthermore, Poisson's ratio is an indicator for determining the difference between gas and fluid saturated materials (Gregory, 1976; Pasquet et al., 2016) and has been shown to be useful to track pressure changes (Prasad, 2002), map the water table depth (Bachrach et al., 2000; Pasquet et al., 2015b; Salem, 2000; Uyanık, 2011), and
200 differentiate gas and fluid in hydrothermal systems (Pasquet et al., 2016). To image the water table with Poisson's ratio, the conceptual model of the geology must be simplified (i.e. no lateral changes) and requires that there are at least a few meters of unsaturated sediments overlying the saturated region to generate a vertical contrast in elastic properties; our study site satisfies both these conditions.

### 3.5 Nuclear Magnetic Resonance

205 Nuclear magnetic resonance (NMR) capitalizes on the existence of a measurable magnetic moment produced by the rotation of hydrogen protons contained in water molecules. At equilibrium, the direction of the magnetic moment points in the direction of a background magnetic field. An NMR measurement emits an electromagnetic pulse at a specific frequency (called Larmor frequency) in order to force protons out of equilibrium. When the excitation pulse ends, the protons return to equilibrium in a process called relaxation. During relaxation, a measurable resonating magnetic moment that decays
210 exponentially can be measured (Bloch, 1946; Brownstein and Tarr, 1979; Torrey, 1956). The initial magnitude of the signal is directly proportional to the number of protons excited, which in near-surface exploration come mostly from groundwater, and the rate of decay (i.e. the relaxation time $T_2$) is related to the pore size. Thus, NMR has the ability to directly measure the amount of groundwater within its measurement volume. For a thorough review of NMR theory, the reader is referred to (Behroozmand et al., 2015) and textbooks dedicated to the theory of NMR (Coates et al., 1999; Dunn et al., 2002; Levitt,
215 2001).

  The decay rate, described by $T_2$, is a function of two distinct processes: the bulk relaxation and the surface relaxation (Brownstein and Tarr, 1979; Cohen and Mendelson, 1982; Grunewald and Knight, 2012). The surface relaxation is controlled by an intrinsic property called the surface relaxivity and the surface-to-pore volume ratio. Surface relaxivity describes a material's ability to intensify relaxation. The dependence on the surface-to-pore volume ratio is what relates the NMR decay
220 to the pore scale properties. In general, materials with larger pores spaces have longer $T_2$ relaxation times (e.g. gravels) and

materials with smaller pores have shorter $T_2$ (e.g. clays). When high quality data is acquired, such as with downhole systems, the $T_2$ relaxation times can be fit using multi-exponential decay curves. The distribution of decay times represents the properties of all the pores within the excited volume. We acquired downhole NMR measurements at 0.25 m depth intervals down a 7.5 m drillhole using a Dart system (Vista Clara). The Dart quantifies water content and $T_2$ decay times in two cylindrical shells of varying radii (12.7 and 15.2 cm) within the drillhole.

## 3.6 Transient Electromagnetics (TEM)

The transient electromagnetic method utilizes a transmitting and receiving loop lying on the earth's surface. The TEM method specifically uses a short-transmitted pulse duration and measures the decay amplitude of the vertical component of the electromagnetic field generated by secondary currents as a function of time. The magnitude and decay rate of the vertical electromagnetic field is related to the electrical conductivity of the subsurface beneath the loop. The penetration depth of the method depends on the underlying conductivity structure and the size of the transmitting loop and the amplitude of the transmitted signal. For a more thorough description of the TEM technique see Telford (1976), Parasnis (1986), or Reynolds (2011).

We collected the TEM data using a Zonge Engineering NanoTEM system. The NanoTEM is a low-power, fast-sampling time-domain TEM system that was specifically designed to provide high resolution images of the near-surface (~50 metres depth). The NanoTEM data were collected using a 20 m x 20 m square transmitter loop with a 5 m x 5 m, single-turn receiving loop. The transmitter coil had an output current of 2 A and a turnoff of ~ 2 µs. The receiving loop sampled at 625 kHz, stacking 256 cycles at a repetition rate of 32 Hz. The stacks were averaged and then inspected to remove noisy data in the late times. The NanoTEM data were inverted using the Aarhusinv program, run using "smooth model" settings (Auken et al., 2006, 2015). The 1D inversion assumes laterally homogeneous layers. All NanoTEM soundings were inverted separately (i.e. there were no lateral constraints,) and placed next to one another and interpolated to generate pseudo 2D profiles of bulk electrical conductivity. The quality of the inversion is determined by a misfit value between the observed and modelled voltages.

## 3.7 Drillhole Soil Sample Measurements

Soil samples were collected at 0.25 m intervals from the continuous core that was retrieved during the shallow drilling program. Each soil sample was placed into an air-tight plastic container to prevent moisture loss and preserved for later analysis in the laboratory for gravimetric water contents and soil pore water salinity. The gravimetric water content was determined as the water loss between the wet and dry sample after three days in an oven at 40 degrees, using standard methodologies as described in Rayment and Higginson (1992). Salinity (i.e. electrical conductivity) of the pore water was measured using a 1:5 mass ratio by combining 20 g of soil and 100 g of ultra-pure water (He et al., 2012, p.5; Rayment and Higginson, 1992). The samples were agitated by rotating in a tumbler device for 48 hours, left to settle for one hour and then an electrical conductivity probe was used to measure the electrical conductivity of the supernatant. The soil water conductivity was determined using

the 1:5 ratio dilution factor. For the shallow drillhole, we have gravimetric water content and soil conductivity as a function of depth. To make comparisons with the NMR data, the gravimetric water content was converted into volumetric water content by multiplying an assumed soil density between 1.3-1.5 g·cm-3. We also assumed the density of water equal to 1 g·cm-3.

## 4 Results

### 4.1 Seismic Results

We use the P-wave profile generated by travel-time tomography (Figure 2a), the S-wave profile estimated through the inversion of surface waves (Figure 2b) to create a Poisson's ratio profile (Eq. 1) (Figure 2c). Under the assumption that changes in Poison's ratio are due to saturation and not changes in lithology, the Poisson's ratio should increase to values close to 0.5 as saturation approaches 100%. In our data, the Poisson's ratio increases with depth and averages out to a value of ~0.46 below an elevation of 5 m a.m.s.l. (Figure 2c). An anomaly occurs between 60 to 80 m along the profile and is the only location where high Poisson's ratios (> 0.4) reach the surface. This observation is not surprising given that this profile is driven by P-wave and S-wave profiles where at 60-80 meters along the profile we observed a drop in S-wave velocities, while the P-wave velocities increased slightly (Figures 2a and 2b). Because the difference in P-wave and S-wave velocity is larger, the Poisson's ratio is also larger (Eq. 1).

The P-wave velocity profile is characterized by two features. The first feature is a laterally homogeneous layer defined by a consistent increase in velocity from about 0.3 km·s-1 to 1.5 km·s-1. The bottom of the feature is defined by a velocity of ~1.5 km·s-1 and corresponds to the depth where the vertical velocity gradient weakens significantly (Figure S3d). This boundary, which is clearly identified in the travel-time picks (Figure S2), defines the bottom of an approximately 13-m-thick horizontal layer at around 0 m elevation (Figure 2a). The second feature is more subtle and is associated with a change of slope in the travel-time picks between 60 and 80 m (Figure S2a). Because of the high quality of the seismic data, the inversion was able to adjust this change in slope in the travel-time curves (Figure S2), which is reflected in the final model (Figure 2a).

Like the P-wave velocity profile, the S-wave profile is laterally continuous (Figure 2b). On average the S-wave velocity increases from 0.2 km·s-1 at the surface to 0.4 km·s-1 in the deepest parts of the model. There is an abrupt increase in velocity around 0 m elevation (Figure 2b), which is consistent with the large change in velocity observed in the P-wave velocity profile (Figure 2a). There is one notable difference between the two profiles occurring between 60 and 80 m., approximately at the same location where we observed the subtle increase in P-wave velocities; unlike the P-wave velocities, the S-wave velocities are defined by a slight decrease in velocity (Figure 2b) which was also a clear and observable feature in the picked dispersion curves (Figure S4).

## 4.2 TEM Results

The 26 NanoTEM soundings show consistency between the soundings (Figure 3a). To ease comparisons to both the S-wave and P-wave profiles, the soundings were interpolated to a 2.5 x 0.5 m grid. In this grid the distance along the x-axis is relative to the start of the seismic profile (Figure 1). The interpolation was done using an adjustable tension continuous curvature spline (Smith and Wessel, 1990). In the interpolated section (Figure 3b), the most resistive feature (< 200 mS·m-1) occurs at the ground surface and extends to an elevation of 10 m above mean sea level (m a.m.s.l.) between 60 to 80 m along the profile. The resistive feature is well constrained by individual soundings (Figure 3a) and extends both laterally and at depth on both sides of the depression between 10 and 5 m a.m.s.l. and from 40 to 160 m along the profile (Figure 3b).

## 4.3 NMR and Soil Sample Results

The downhole NMR results show that the volumetric water contents of the soil profile vary between 0 and 0.25 m3/m3, with a gradual increase with depth (Figure 4a). The maximum water content of 0.25 m3/m3 was measured between 6.75 and 7 m depth, consistent with the measured water level depth (6.8 m) (Figure 4a). The average amplitude of the noise in the water contents determined by NMR is ~0.05 m3/m3. Therefore, inverted water contents less than 0.05 m3/m3 are less reliable. Signals from the soundings can be found in the supplementary material (Figure S6). Above the water table, the NMR data showed a rise in water contents above 0.05 m3/m3 between 1.75 and 3 m depth. The water within this region contains low $T_2$ relaxation times (< 0.01 s). A similar pattern in water content and $T_2$ decay times occurs between 4 and 6 m depth (Figure 4a and 4b). Around 6 m depth the $T_2$ distributions transition from shorter to longer (>0.01 s) relaxation times and the water contents also increase (Figure 4b). This gradual increase in the $T_2$ decay times and water content is likely to be the capillary fringe, where the remaining pore space fills from smallest to largest pores. At depths below the measured water level (6.8 m), the $T_2$ distributions normalize and have a value just over 0.01 s, which is consistent with clays.

The gravimetric water content measured from the drill hole core samples had an average of 0.15 with a standard deviation of 0.02 and showed very little variation with depth (Figure 4c). The soil pore water conductivity also showed little variation with depth, having an average conductivity of 1,123 μS·cm-1 with a standard deviation of 424 μS·cm-1 (Figure 4c). When converted to mSm-1 (112.3 ± 42.5 mSm-1) the soil conductivities are in within the range of conductivities observed in the TEM profile. In contrast to soil conductivities, the measured groundwater conductivity was 14,750 μS·cm-1 (conductivity of sea water is ~50,000 μS·cm-1). The TEM conductivity represents an average between the soil conductivity and groundwater conductivity, which is why we do not observe high conductivity values below the water table (Figure 3). There is also notable difference in the gravimetric water contents and NMR water contents (Figure 3). This difference could be attributed to the fact that the NMR cannot detect water in the smallest pores. Below the water table, where the assumption of fully saturated pores is likely satisfied, the gravimetric water contents and NMR water contents are in much closer agreement (Figure 4a).

**5 Discussion**

 **5.1 Geophysically Inferred Water Table Depth**

To estimate a value of Poisson's ratio that represents the water table, we laterally averaged two regions along the profile to produce two 1D profiles with standard deviations. The standard deviations represent the lateral variability. The first laterally averaged region was between 60-80 m, where the large anomaly occurs and where higher values of Poisson's ratio reach the surface (Figure 5a). The second region was chosen to be from 120 to 220 m because qualitatively it appears laterally 320 uniform and includes the drillhole location (drillhole located at 220 m). These two averaged 1D profiles, when plotted side-by-side, show a similar trend of increasing Poisson's ratio (Figure 5c) but present a clear offset. Near the surface the difference is largest, but the two curves begin to converge near the water level measurement of 6.8 m depth below ground and the highest water contents from the NMR (Figure 5c). At the inferred water table, the values of Poisson's ratio between 60-80 m and 120-220 m are $0.454 \pm 0.004$ and $0.475 \pm 0.002$, respectively (Figure 5b). Here we use a value of 0.46 as the contour that represents 325 the water table, which we refer to as the geophysically inferred water table depth. The value of 0.45 also validated against the manual water level measurements (6.8 m depth below ground) and the downhole NMR water content profile from the drill hole (occurring at 220 m along the profile). It also corresponds well with previous values given by Pasquet et al. (2015a, b).

Under the assumption of a flat-water table from the drillhole, the contour value of 0.46 matches qualitatively the depth to water between 0-60 m and again between 80 and 220 m (Figure 5a). There is one notable deviation occurring between 330 60-80 m where we highlight anomalies in all three geophysical methods. We observed a slight increase in P-wave velocities (Figure 2a), slightly lower S-wave velocities (Figure 2b), and a resistive feature in the NanoTEM data (Figure 3). As a result, the geophysically inferred water table depth at this location along the profile differs from the manually measured water level (Figure 6a). We interpret this rise in Poisson's ratios as the water table rising toward the surface beneath the subtle topographic depression in the landscape, representing the ephemeral drainage feature (Figure 6b).

Using a contour value of 0.46 provides an estimate for the water table, but the boundary is fuzzy and possibly transitional (Figure 5a). The fuzziness of the boundary could be explained by two processes. First, partial saturation could be occurring above the water table. Second, the water table boundary could be well defined, but it is smoothed over by the geophysical inversions. The smoothing is difficult to quantify and is complicated by the fact that the P-wave and S-wave velocities come from two different inversions based on different physics. More research is needed to understand and compare 340 the sample volumes of the travel-time tomography and surface wave inversions. The first interpretation of a transitional zone between unsaturated and saturated sediments is more likely because of observations in the NMR data (Figure 4) and the presence of water measured in the collected soil samples (Figure 5c).

In the downhole NMR data, we are confident with measured water contents greater than $0.05$ m$^3$/m$^3$. At 4 m depth the water content is well above $0.05$ m$^3$/m$^3$ and shows a linear increase until a maximum value of $0.25$ m$^3$/m$^3$ is reached 345 between 6.75 and 7.0 m below the surface (Figure 4). Below the water table, the maximum water content likely represents total porosity. All the NMR responses above the water table have low $T_2$ decay times (Figure 4b), which can either be indicative

of clay or caused by small pores, which preferentially fill and hold water after being drained (i.e. leading to partial saturation of the medium) (Walsh et al., 2014). The preferentially filled pores seems more likely because we know that the measurements were made within the vadose zone and the measured gravimetric water contents of the drillhole core showed that samples retained some moisture. The most important observation provided by the NMR data is that partially saturated sediments exist at least 3 m above the water table (Figure 5c). This partially saturated region of the soil profile will likely increase the Poisson's ratios and provides an explanation for the transitional and fuzzy boundary we observe in the seismic data. Furthermore, if the 0.46 contour is shifted upward 3 m based on the NMR observation, it qualitatively matches the point where the Poisson's ratio begins to increase (Figure 5a). From the combined interpretation of seismic data, manual water level measurement, and NMR data, we are therefore able to identify a mound in the water table underneath the small topographic depression, existing between 60-80 m along the profile (Figure 5b). The NMR data and Poisson's ratio suggest the existence of a ~3 m thick section of partially saturated sediments on top of the water table along the profile (Figure 5b).

## 5.2 Geophysically Identified Recharge Processes

In the previous section we relied heavily on the seismic data, NMR data, soil cores and water level to define a geophysically inferred water table depth along the study transect (Figure 5b). We argued for the existence of a 3 m thick partially saturated region above the water table based on water contents from the NMR data (Figure 5c). In this section we utilize the bulk electrical conductivities obtained from the NanoTEM data to strengthen the interpretation that the Poisson's ratio anomaly between 60 to 80 m along the transect is caused by an increase in saturation to demonstrate that the subtle topographic surface depression acts as a localized recharge zone.

Ambiguities exist in geophysical measurements because they measure physical properties that are related to the processes that we are trying to understand. Although we don't expect any lithological variation based on the known soil and geological mapping of the area, it is possible that the region of high Poisson's ratios is a result of higher clay content, since materials that are deformed easily will have higher Poisson's ratios. The Poisson's ratio for pure quartz, a stiff mineral, is between 0.06 and 0.08, Kaolinite is 0.14, and clays are around 0.34 and 0.35 (Mavko et al., 2009). It would be reasonable to assume higher clay content as an alternative interpretation to explain the higher Poisson's ratios under the topographic surface depression. Here the conductivities from the NanoTEM provide evidence to suggest that an increase in clay content is unlikely. If the high Poisson's ratio were due to an increased clay content, we would expect the electrical conductivities to rise—but we observe the opposite. The subsurface is more resistive at the location where the Poisson's ratios rise.

Underneath the small depression in topography between 60-80 m along the seismic profile we have anomalies in all three geophysical data sets: 1) the P-wave velocities increase, 2) the S-wave velocities decrease, and 3) the electrical conductivities decrease. As discussed in the previous section, the first two anomalies cause the Poisson's ratio to rise, which we interpret as a rise of the water table, or at a minimum increase in water saturation. Here we believe the decrease in electrical conductivity is the result of more conductive groundwater being replaced by fresher water that has infiltrated from rainfall events. Although we did not measure the electrical conductivity of rainwater, we know that the groundwater from the shallow

aquifer at the study site has a much higher conductivity (14,750 µS·cm-1) which is similar to observed salinities in the shallow quaternary system across the NAP, which range between 2000 and 13,000 mg·L-1 (3,075 to 20,000 µS·cm-1 ) (Department for Water, 2010). Typically, an increase in saturation causes an increase in electrical conductivity. A simplified and general modelling exercise using Archie's Law shows that if we replace the water in the pores with a more resistive fluid, it is possible to get a drop in electrical conductivity even if the saturation is increased (refer to Supplementary Material). Thus the decrease

in electrical conductivity supports the interpretation that the rise in Poisson's ratios is caused by an increase in water content and not an increase in clay content.

## 5.3 Hydrogeological Implications

We combine all the geophysical observations to construct a hydrogeological interpretation of the study site (Figure 6). First, based on the seismic data and the measured water depth from the nearby drillhole we can identify a rise in the water

table beneath the small topographic depression. It is likely that this rise in water table has a partially saturated region that is ~3 m thick above it. Because of the observed drop in electrical conductivities, we interpret this feature as a saturation increase and not a change due to an increase in clay content. We interpret the drop in electrical conductivities as fresher water diluting and mixing with the ambient saline groundwater of the Quaternary aquifer. The resistive feature lies above the partially saturated or saturated zones between 80 and 100 m and again between 120-140 m (Figure 6).

Our study has shown that the smaller tributaries and ephemeral streams across the lower lying areas of the Adelaide Plains are acting as localized sources for recharge to the Quaternary and Tertiary aquifers. This suggests that there is localized areas of recharge across the NAP associated with these types of subtle features—an interpretation that is consistent with the findings of Bresciani et al. (2018), which showed that the main recharge mechanisms to the Quaternary and Tertiary aquifer systems across the Adelaide Plains was through surface water infiltration along the large river systems (e.g the Gawler and

Little Para rivers) that have their headwaters in the Mount Lofty Ranges and outlets towards the coast.

It should be noted that our hydrogeological interpretation is based on a single snapshot in time. Without time-lapse geophysical measurements, groundwater samples taken from within the groundwater mound and either side, or long-term monitoring of groundwater observations wells, it is not possible to definitively quantify the recharge rates in these systems. Nor is it possible to determine if the groundwater mound is a result of a recent rainfall event or if it is a more stable feature. It

seems reasonable, given the evidence of ephemeral surface drainage features in the LiDAR data (Figure 1) and the high clay content of the near-subsurface that surface water would flow towards subtle depressions in the landscape and eventually out to St Vincent Gulf (Figure 6). These small ephemeral features are unmonitored, so it is unknown how quickly or how much water flows through them during storm events. The NAP is topographically flat so it is possible that instead of surface water flowing out towards the ocean, water might accumulate in these low-lying features after large rainfall or storm events and

gradually infiltrate over longer periods of time. The ponded water from such rainfall events would produce localized recharge to the underlying aquifer system (Figure 6). The recharge water would be fresher than the groundwater already in the Quaternary aquifer system.

The hydrological conceptualization based on the geophysical data (Figure 6) could be confirmed or rejected by drilling and sampling the groundwater via an additional shallow drillhole across the shallow topographic depression. The combination of geophysical data has provided a new perspective that allows us to speculate about important local hydrological processes taking place in the NAP. Furthermore, the combined geophysical approach presented here can be used to guide and plan more widespread investigations focused on understanding the role of these subtle ephemeral features across this flat semi-arid landscape. The combined geophysical approach provided a vital conceptual framework for the hydrological processes occurring within the area. Future work should focus on combining the geophysical measurements with more traditional hydrological and geochemical measurements to fully explore and test the hydrogeological conceptualization suggested in this manuscript and the transient nature of the recharge mechanisms (Figure 6).

## 5.4 Applying the Combined Geophysical Approach to other Semi-arid Regions

Throughout this study we used well established geophysical methods. Each of these methods have open source inversions available or the equipment comes with easy to use inversion software. Thus, there is nothing novel about the processing of each individual geophysical dataset, but the combination of all these methods can be used to rapidly explore flat lying features to test a given conceptual model of the recharge processes in a flat lying semi-arid landscape. In order to facilitate and expand the use of this combined geophysical approach to other semi-arid streams or features, we highlight some of the uncertainty, limitations, advantages, and critical assumptions that went into building the hydrological conceptualization so that this methodology might be transferred to other semi-arid areas that are common around the world.

We relied heavily on the manual water level measurement and downhole NMR data. The geophysical mapped water table essentially extended from the water level that we were able to measure at the drillhole location. The drillhole data were critical to calibrate the value of Poisson's ratio that we used to represent the water table. The method would be much more powerful if the drillhole was not required, but because this was the first survey of its kind in the region we needed to confirm where the water level was to interpret the Poisson's ratio. Now, with value of 0.45 it would be possible to run a survey without the drillhole and predict the water level without a drillhole. Thus, some validation is required prior to extending the methodologies throughout the NAP.

The NAP provided ideal conditions for us to exploit Poisson's ratio to map the water table in detail. The NAP was ideal because the subsurface was broadly homogeneous, and there were no abrupt or lateral variations in the lithology. Lithological variation would complicate the interpretation of the Poisson's ratio because all the changes could not be attributed solely to changes in saturation. We were also specific in selecting a location where the water table was between 3 and 10 m depth. In order to image the saturated zone with seismic methodologies, we required an elastic contrast between the unsaturated and saturated zones. Although uncertainty is difficult to quantify given the different sample volumes and wavelengths of the seismic wave field and Rayleigh waves (work that extends beyond the scope of this paper) we believe that having at least three meters of unsaturated zone above the water table should provide a strong enough contrast to image. Furthermore, the inversion of surface waves is limited by the frequency content of the source and the peak geophone frequency. In our case, with 14 Hz

geophones, imaging a water table that is below ~10-12 meters would be difficult. Thus, to improve chances of success, the seismic approach should be applied in regions where the water table is between 3-10 m in a homogeneous material. We used the existing 47 boreholes from another study (Hatch et al., 2019) to select a site that satisfied this condition.

The additional information provided by the NanoTEM data helped reduce ambiguities observed in the Poisson's ratio profile. Without this additional information it would have been difficult to determine if the anomaly was caused by an increased clay content or an increase in water content. Thus, the electrical conductivity data was critical to our hydrological interpretation. It should be noted that the NanoTEM data could also be replaced by another independent observation e.g. other near surface geophysical methods or soil conductivity profiles at several locations along the transect. Regardless, more observational evidence, even if they are point measurements, will aid in the interpretation of the geophysical images.

## 455  6 Conclusions

      We have shown that the combination of P-wave and S-wave velocities, electrical conductivities, and surface NMR can identify small-scale ephemeral recharge features in a semi-arid landscape without the use of time-lapse measurements. The seismic data were used to calculate Poisson's ratio and served as the foundation to geophysically infer the water table depth. The NMR data showed a 3 m thick region of partially saturated sediments, and the electrical conductivities from the

460 NanoTEM provided additional evidence to support an increase in water content opposed to an increase in clay content. The combination of all four data sets has provided a hydrogeological framework where we are observing fresher water recharging and replenishing the underlying saline Quaternary aquifer system. Although the timing or flux rates of the recharge cannot be determined with our data, we have shown that small scale ephemeral features could play a vital role in recharge mechanisms to the shallow unconfined aquifers of the low-lying semi-arid landscape of the Northern Adelaide Plains. The interpretation of

465 the geophysical data still requires more traditional hydrogeological measurements to completely validate the results, but we have demonstrated that the spatially exhaustive perspective gained, using near-surface geophysical methods can be valuable to understanding the recharge processes and conceptualization of semi-arid hydrological systems.

## 7 Acknowledgments

We would like to thank Adrian Costar for help in collection of the seismic data set. We acknowledge funding from the Goyder

Institute for Water Research for the project ED-17-01, 'Sustainable expansion of irrigated agriculture and horticulture in Northern Adelaide Corridor: Task 4 – assessment of depth to groundwater (proof of concept)'. We are grateful for the land access provided by John Gordon. We would also like to thank Chris Li and Sebastian Lamontagne for providing valuable comments that improved the manuscript. We would also like to thank Alan MacDonald and Kent Inverarity for taking the time to review this manuscript and provide comments that helped improve the overall clarity of the manuscript.

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

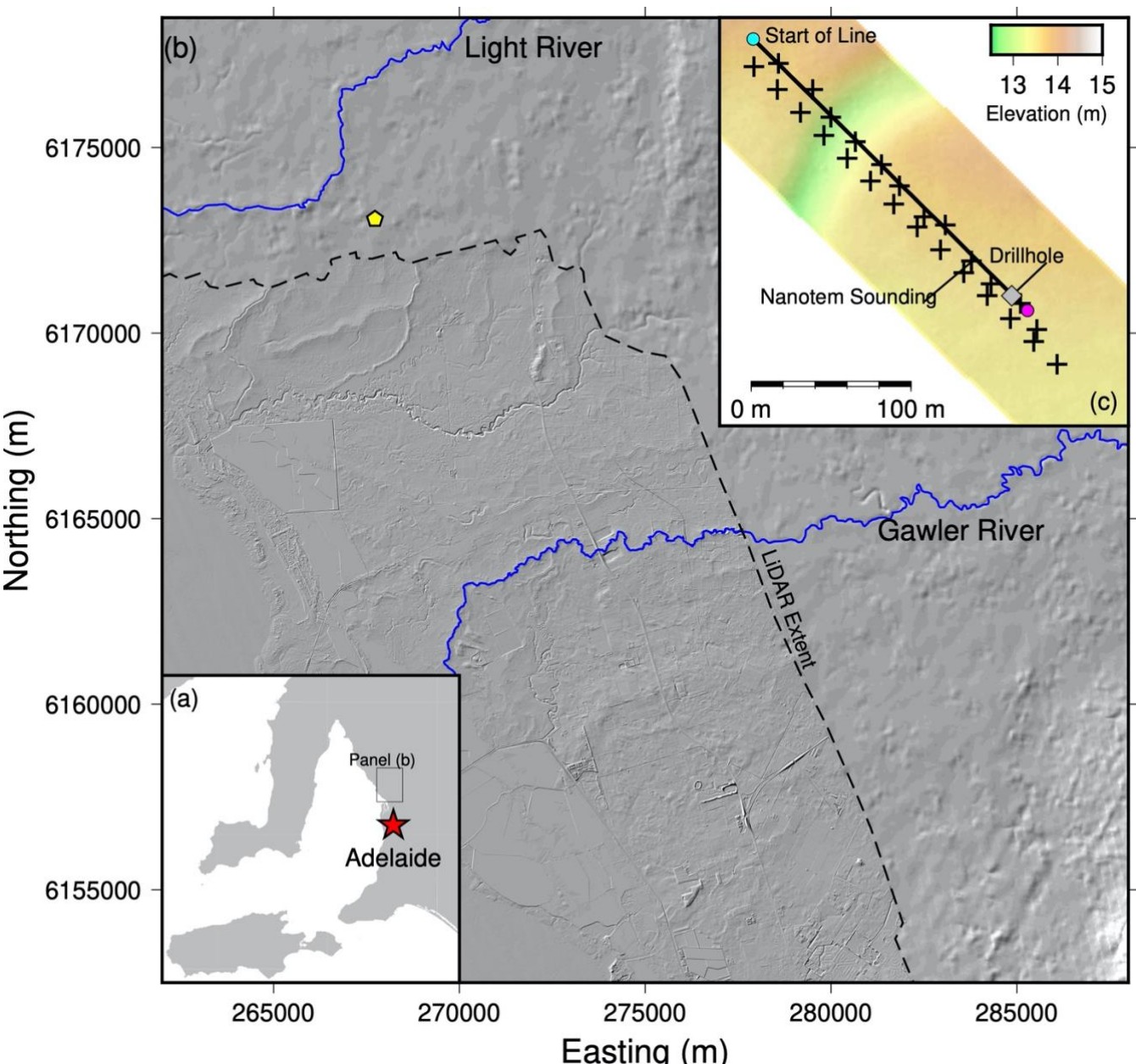

**Figure 1: (a) Inset map showing the general location of the study area relative to the city of Adelaide, South Australia. (b) Hillshaded topographic relief with the LiDAR data overlaid. The northern extent of high-resolution LiDAR DEM data is marked with a dashed line. The yellow pentagon represents the site location. The Light and Gawler Rivers are shown in blue. (c) High resolution topography (drone based) of the field area where the geophysical testing took place. The thick black line is the seismic line, where the cyan dot is the start and the magenta dot is the end. Black X's represent the location of NanoTEM soundings. The grey square is the shallow drillhole location and also where the downhole NMR data were collected.**

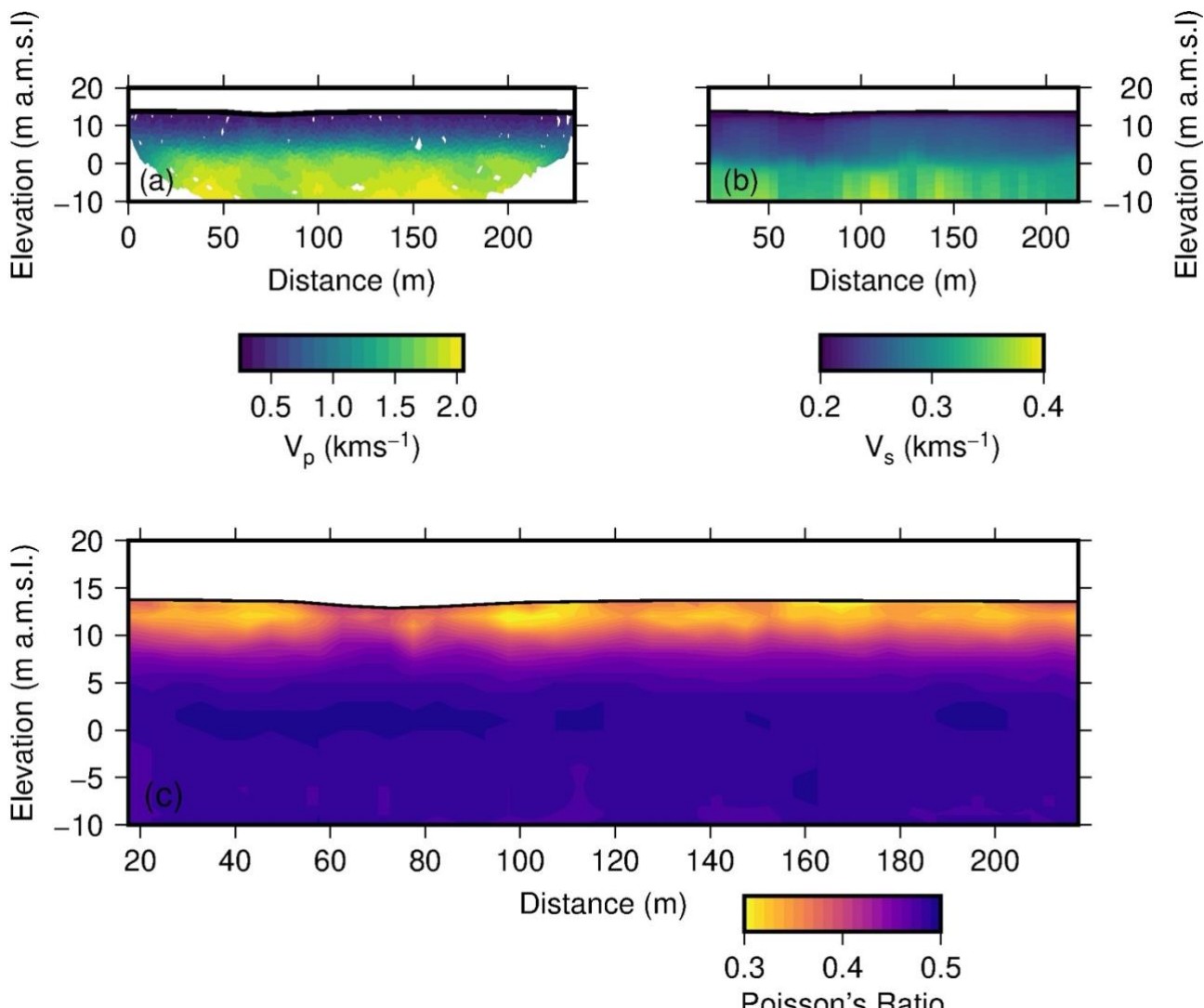

Figure 2. (a) The P-wave velocity results shown at 2x vertical exaggeration. Areas where no rays pass through a model cell have been masked out. (b) The S-wave velocity results shown at 2x vertical exaggeration. This image shows the 42 1D inversions side-by-side; no interpolation has been applied. (c) Profile of Poisson's ratio calculated using Equation 1 and the profiles shown in panels (a) and (b).

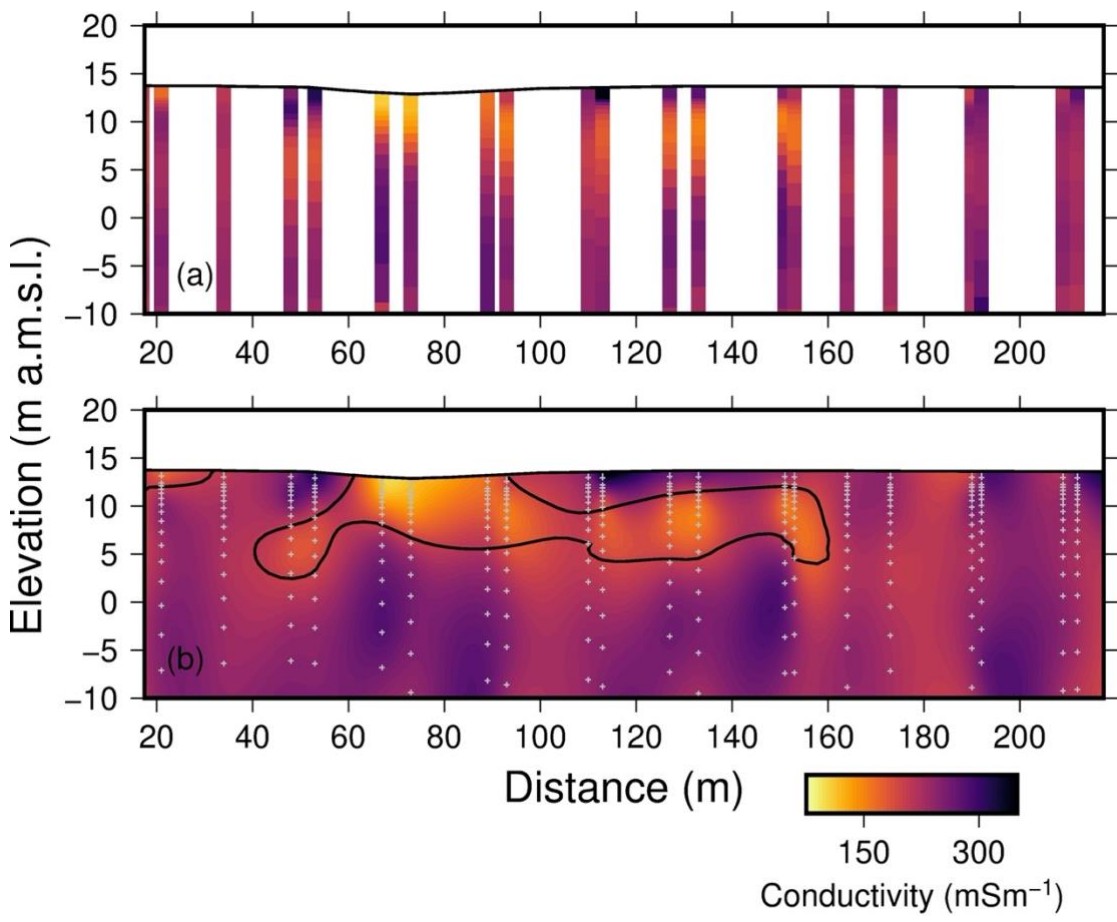

**Figure 3. Electrical conductivity results produced by the NanoTEM soundings. The x-axis is distance from the first geophone (Figure 1). Both panels are shown at 2x vertical exaggeration. (a) 1D conductivity profiles plotted at their inverted resolutions as well as their spatial locations. (b) The interpolated conductivity section. The black contour represents 200 mS·m-1. The small grey plus symbols represent the data used for the interpolation.**

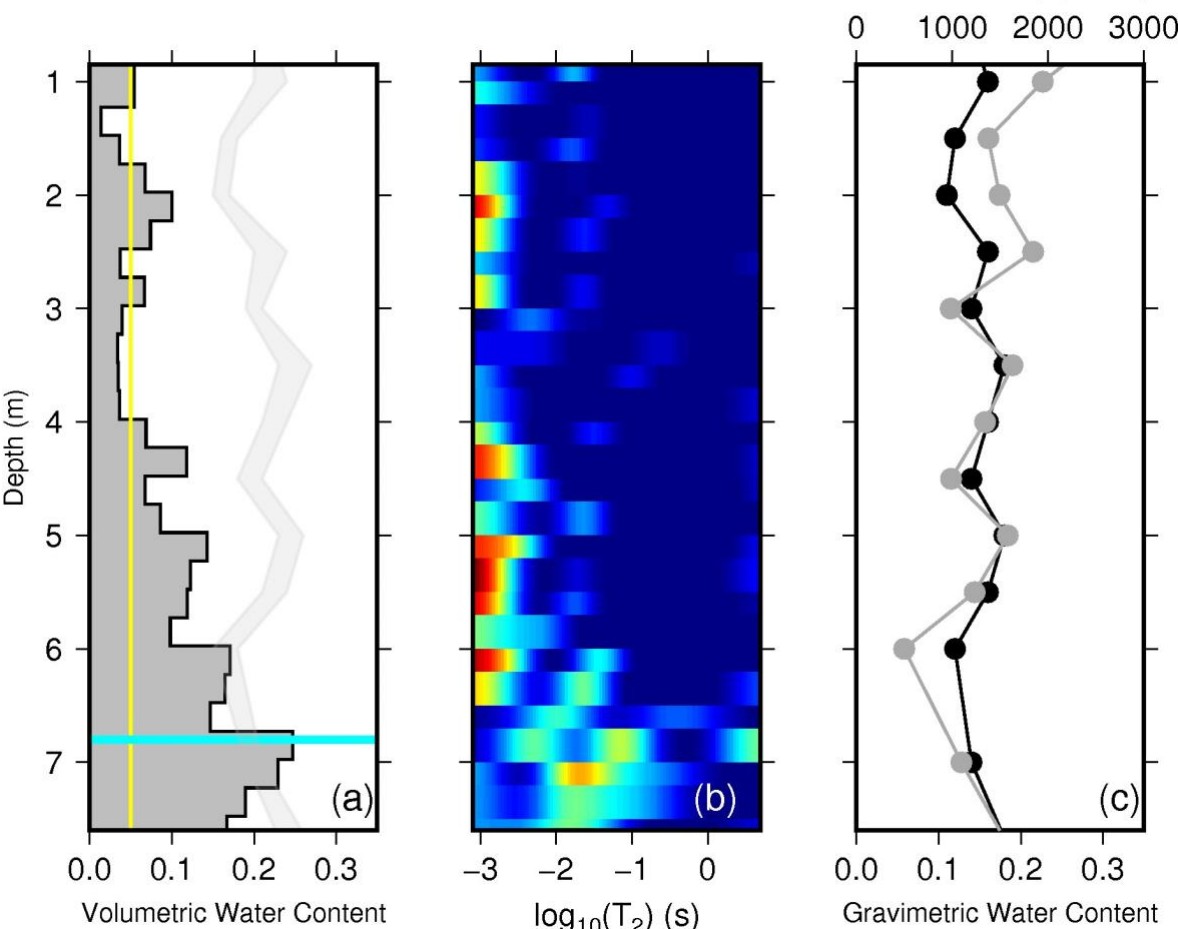

**Figure 4. Results from the downhole NMR sounding and soil samples at the drillhole location versus depth (metres below ground level (Figure 1). (a) The water content profile from the downhole NMR data. The thin vertical yellow line shows the average noise level (0.05 m₃/m₃) below which water content estimates are questionable. The thick horizontal cyan line represents the manually measured water level from ground surface (6.8 m). The thin transparent region is the volumetric water content estimated from the measured gravimetric water content, assuming a soil density between 1.3 and 1.5 g cm₃. (b) The T₂ distributions that produced the water content curves in panel (a). The maximum water contents are calculated by summing the area under the distribution. (c) Soil conductivity (grey) and gravimetric water contents (black) as a function of depth.**

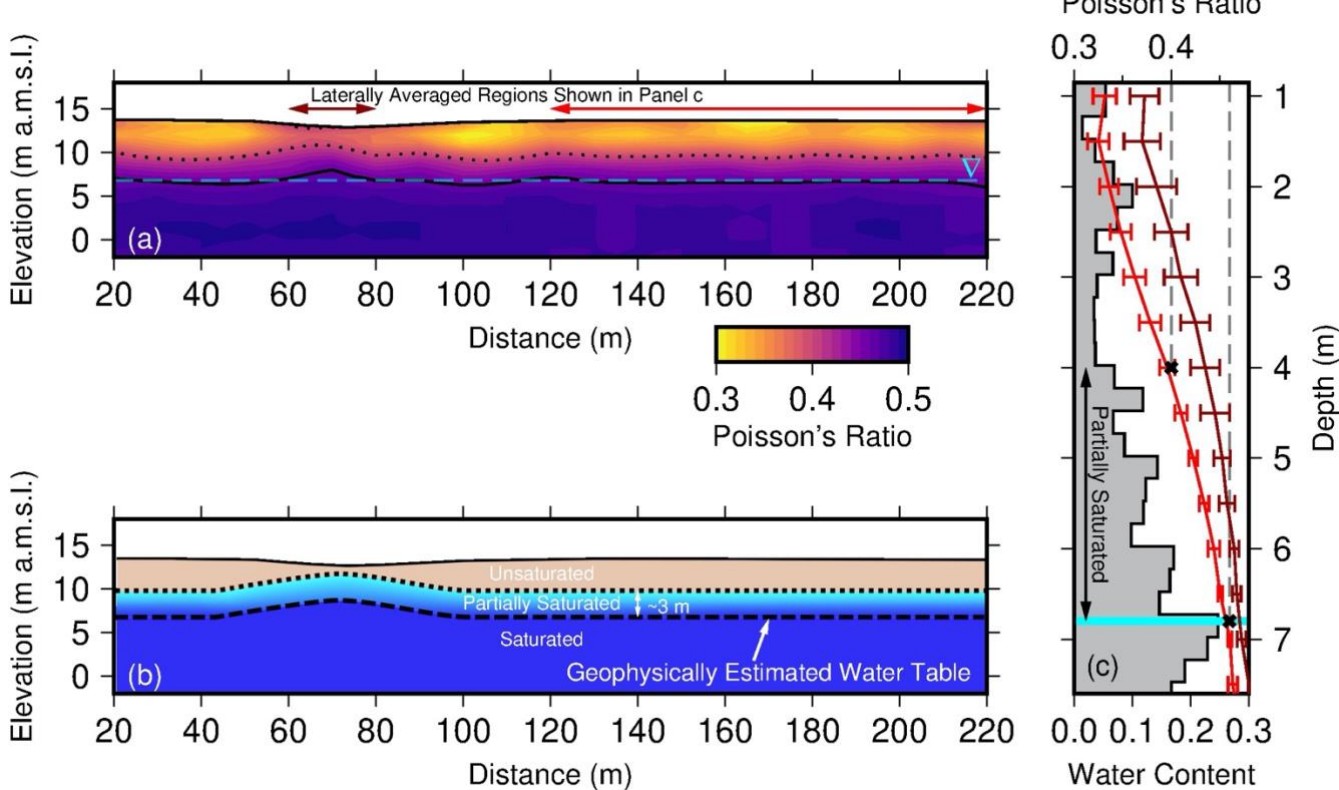

Figure 5 (a) Profile of Poisson's ratio calculated using Equation 1 and the profiles shown in Figures 2 and 3. The solid black contour represents a Poisson's ratio of 0.46. The dashed cyan line is the depth of water measured at the drill hole located at 220 m (6.8 m below ground surface) and extends horizontally across the profile to be able to visualise the changes beneath the ephemeral feature. The dotted contour line is the 0.4 contour line, which is consistent with a 3 m thick partially saturated region. (b) Geophysically interpreted hydrogeological cross-section. The unsaturated zone is quantified by areas with Poisson's ratios less than 0.40. The partially saturated region, with a thickness of approximately 3 m (determined from NMR in panel (c)) has Poisson's ratio between 0.4 and 0.46. The fully saturated region has Poisson's ratios greater than 0.46. The geophysically inferred water table is approximated by the 0.46 contour in panel (a). (c) The water content profile from Figure 4. The partially saturated region from 4 to 6.8 m depth is highlighted. The horizontal cyan bar is the manual water level measurement (6.8 m). Overlain on the water content profile are the two horizontally averaged 1D Poisson's ratio profiles. The red line is averaged from 120-220 m and the maroon line is averaged between 60 and 80 m along the profile. Black dashed lines and solid crosses highlight the Poisson's ratio contour values of 0.4 and 0.46 chosen in panel (a).

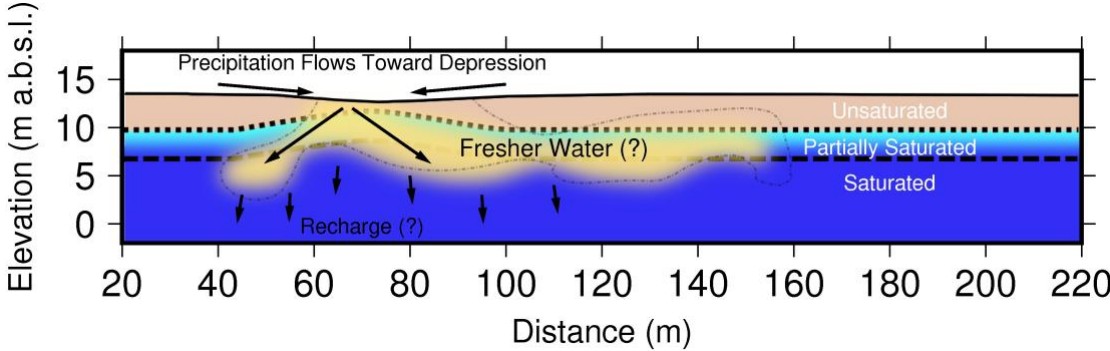

**Figure 6. The final hydrogeological framework for a subtle ephemeral surface water feature in the semi-arid landscape of the Northern Adelaide Plains. The underlying map is based on the seismic data (Figure 6b). The yellow region is our interpreted region of fresher water that has recharged the shallow unconfined aquifer system. Here water flows across the land surface and collects into the subtle ephemeral drainage feature. From there water is recharged into the underlying aquifer system and the hydraulic gradient drives recharging water away from the groundwater mound, to either side. During the recharge process, the fresher recharge water is mixing with the ambient saline groundwater of the shallow Quaternary aquifer.**

Dear Dr. MacDonald,

Thank you for taking the time to read and assess our manuscript. You have provided some excellent advice that will help improve the overall readability to a general hydrological audience; which was the intent despite being a geophysical dense manuscript. In this response I will address your comments.

**General Comments**

You highlighted six general comments. All six of these comments are beneficial if implemented and they will not change the scientific data or interpretation instead they will greatly improve readability. In this section I will summarise your general comments and then provide details on how the manuscript will be changed.

1. You suggest that we should discuss and frame the manuscript with the Poisson's ratio (rather than P and S Waves) – particularly in the abstract and results. Too much time is spent on individual interpretation of P and S wave data.

*First and foremost, it's important that Poisson's ratio be mentioned in the abstract, this is an easy fix and provides the readers a better overview of the data contained within the manuscript. Second, we agree that the context should be framed with Poisson's ratio but still believe it is important to show and describe the P-wave and S-wave data that the Poisson's Ratio profile is based on. From a geophysical point of view it is critically important to highlight the slight rise in P-wave velocities (shown in the travel-time picks in Supplementary Material) and slight drop in S-wave velocities (shown in the dispersion curves in Supplementary Material) to ensure that the subtle rise in the Poisson's ratio profile is not an artifact. Essentially, the calculation of Poisson's ratio is routine but the collection of our P-wave and S-wave data in the same survey is not (see Pasquet et al., 2017). The MASW is traditionally done in a separate survey. Thus, the P-wave and S-wave data come from completely separate inversions (and physics for that matter). This is another reason that the reader needs to see and read our description of the P-Wave and S-wave profiles.*

*To address this comment, we will do as you suggested later on in the manuscript and combine Figures 2 and 3. In this new figure we would add Poisson's ratio. In this new Figure we will make the Poisson's Ratio profile double the size of the P-wave and S-wave profiles. This seems to be a good compromise of adding emphasis to Poisson's Ratio but also retaining the P-wave and S-wave profiles that the Poisson's Ratio profile is constructed on. We will alter the results section and open with the description of Poisson's Ratio and leave the description of the P-wave and S-wave profiles in two subsequent paragraphs. Again, that way we are stressing the Poisson's ratio profile but the descriptions and data for P-wave and S-wave profiles remain.*

2.  You suggest that we provide more explanation to understanding the difference between the in-situ data (gravitation water content, soil water conductivity) and the geophysical data.

*As the reviewer correctly points out we spent a lot more time on the NMR results at the drillhole and limited text was provided on the differences between the NMR water contents and the gravimetric water contents. To address this comment, we would break the single paragraph in section 4.3 into two paragraphs. The second paragraph will now explicitly point out that the soil conductivities are in the same range as the TEM conductivities. It will also be explicitly said that the groundwater is conductive but the TEM is measuring an average of the soil and groundwater conductivities so that's why we are not seeing the high conductivities that were measured in the porewater. Lastly, we will add a sentence stating that the NMR cannot measure all of the water in the smallest pores. Since we are clearly dealing with a clay, visible in the logging core and by the short T2 decay times, it is not surprising that the gravimetric water contents are higher above the water table. We suspect that if we had access to a laboratory NMR device that has the ability to measure water in even smaller pores the improvement between that the two methods would improve. We would have been more worried if the NMR water contents were higher than the gravimetric water contents. We are comforted by the fact that below the water table, where all of the pores are assumed to be full, the gravimetric and NMR water contents are in much better agreement.*

3.  Suggest that we delete the section on the use of Archie's equation which is unreliable in this context.

*We agree with this comment. After reading this section again and again the only point that we want the readers to take away from this section is that it is possible that the electrical conductivity will decrease if a more resistive fluid goes into the pore space. This section was a "thought" experiment that helped convince us that this was possible but doesn't really need to be in the main manuscript. We ran this experiment because typically a material will get more conductive as it becomes more saturated. We just wanted to make sure it was possible to decrease the conductivity even under saturating conditions. The other reviewer, Dr. Inverarity, also remarked on this part of the paper and pointed out that we did not thoroughly test a range of literature values and actually suggested we expand the model to potentially predict a probability. That was not the point of the exercise so we would take your advice and move this section to the Supplementary Material. To address this comment in the manuscript we will move Figure 7 to the Supplementary Material and replace two paragraphs of our discussion on the topic with one sentence, "A simplified and general modelling exercise using Archie's Law shows that if we replace the water in the pores with a more resistive fluid, it is possible to get a drop in electrical conductivity even if the saturation is increased (refer to Supplementary Material)."*

4.  You suggest that the wider context and implications are overplayed. Although the paper identifies recharge occurring, it does not indicate its significance to the overall system, and similarly they have not identified a new conceptual model for groundwater recharge.

*We agree. To address this comment, we would point out that the current working conceptual model is that major river systems that have their headwaters in the Mount Lofty Ranges become recharge features (the major recharge mechanism to the underlying aquifer system) as they make their way across the Adelaide Plains (mountain front recharge) as was proposed by Bresciani et al., 2018. We would de-emphasize and remove the idea of competing hydrological conceptualization for the regions. Thus some changes in the introduction section will be made to provide*

*the broader current hydrological framework. In the discussion section (5.3) we will change the wording to say we can extend the ideas of Bresciani et al. (2018) to apply to smaller scale features as well. This way we are no longer proposing a new conceptual model just suggesting that the current hydrological framework might also apply at a smaller scale.*

5.  In general, you suggest that language and diagrams need some improvement. Particularly the overuse of the word "unique" and some colloquial language throughout the text.

*Language like unique and novel did show up too many times in the manuscript and can and will be removed. During*

*the writing process we focused on what was new about our approach to help us stay focused but if the manuscript is written correctly the readers should pick up on this.*

6.  Lastly you point out that we provide a tantalizing glimpse of wider data from 47 research boreholes not included in this study. Are they being interpreted elsewhere? Or could they be used to upscale their results?

*The reviewer is correct that we did show the 47 boreholes in Figure 1. This has clearly had the unintended effect that we will be using all 47 boreholes in the study. To change this in the manuscript we will remove the data from Figure 1 and add a reference to the report that contains the data (Hatch et al., 2019). We will also add a sentence or two in the Geologic Setting section explaining that we used these data to pick the site location. We knew that we would have*

*limited depth penetration with surface wave measurements so we were looking for a site where we know the water table would be between 3-10 m. This assumption is discussed further in the discussion (section 5.4) of the paper as it is a limitation to our approach. The removal of the 47 data points also helps clean up the clutter of Figure 1 as the reviewer suggest later.*

*Hatch, Michael, Okke Batelaan, Eddie Banks, Brady Flinchum, and Megan Hancock. "Sustainable Expansion of Irrigated Agriculture and Horticulture in Northern Adelaide Corridor: Task 4 – Assessment of Depth to Groundwater (Proof of Concept)." Technical Report Series. Goyder Institute for Water Research, 2019.*

**Specific Comments**

•  Abstract – mention the Poisson's ratio – delete references to unique. Line 18. Your results show that localized recharge is occurring, not that all recharge is localized. Also, you don't know how significant this is to the broader system – so change to may play and important role in gw recharge in dryland areas

    *We would follow this advice by adding Poisson's ratio in the abstract and point out that ephemeral features **may** play*
*an important role which would keep us from overplaying the implication early in the manuscript.*

    •  Line 35 – you've missed out Water level fluctuation method. Probably one of the most common on semi-arid areas. You could quote the recent Cuthbert et al 2019 Nature paper

*Sorry we missed this one. We were trying to include as many as possible. We will add a reference to the water table fluctuation method in the Introduction section.*

    •  Paragraph at Line 55 – Not sure you can say that ephemeral stream recharge processes are usually undertaken by time lapse and that there is not a one-off survey method that exists. For example many people have used
groundwater chemistry and environmental tracers (using existing boreholes) to identify that groundwater recharge is occurring. Also people have used ERT to show fresh water over saline.

    *The opening sentence of this paragraph is "ephemeral features are an ideal target for geophysical survey". The intent here was to make sure the reader new this paragraph was going to be all about geophysical measurements and*
*ephemeral recharge. We mentioned calculating recharge via groundwater chemistry and environmental tracers in the paragraph above. To our knowledge a geophysical approach that identifies recharge doesn't really exist without a time-lapse measurement, but this is a broad statement and as you suggest probably not true. This is because we measure geophysical properties such as velocity or electrical conductivity but really want water contents and hydraulic conductivities. This also has to do with the fact that recharge is different than infiltration. To confirm*
*recharge, a confirmation of a change in water table must be observed, hence the application of time-lapse geophysical measurements. Nonetheless, we will weaken this claim and change the last sentence of the paragraph to say, "It is still challenging to find a geophysical approach that can be deployed rapidly (that is without a time-lapse setup) to determine if an ephemeral drainage feature is acting as a groundwater recharge feature."*

•  Paragraph line 66 – Not really a unique combination. Just say a combination. Would strengthen the paper if you discuss and frame with using Poisson's ratio rather than independent S and P wave

*We will follow this reviewer's advice and remove "unique" and "novel" etc.*

• Site description section. Much of what is here is wider context and immaterial. Please reduce this section to just describe the site and local hydrogeology of relevance. Also please mention the vegetation. The 47 boreholes also confused me. Is there a separate paper using these data? I was hoping the paper was going to upscale the results using these boreholes. It also raises the question that most of the information reported in the paper could have been gained from rapidly drilling 10 shallow piezometers across the site to 8 m.


*We agree with the reviewer on this comment and it is stressed in the clutter of the first figure. Major changes to this section will remove the reference to the Tertiary aquifer systems since they are not the focus of this study. We will also remove the 47 boreholes shown on the map so the reader is not misled into thinking we will be using them. The study site description section will be reduced to four shorter paragraphs where P1 is on the overall basin and climate,*

*P2 is specific features relating to NAP mainly that the groundwater is salty, P3 is about the current working conceptual model of recharge mechanisms, and P4 is a description of our focused site. This approach will also simplify Figure 1.*

• Line 300 Soil Sample results. These need much more explanation and are skipped over in the paper. Why are the
gravitational water content and conductivity data so different from the geophysical data? Looking at the plots they could be from a different borehole in another location.

*See the response to general comment number 2 in the section above. We will address this as you pointed out. Though the use of NMR in the vadose zone is still in its early stages and the samples were never processed through a*
*laboratory NMR device, they are not a one to one comparison. However, below the the water table, where pores are assumed to be fully saturated, the difference between gravimetric and NMR water contents are similar. We would have been much more concerned if the NMR water contents where higher than the gravimetric water contents in the vadose zone. Furthermore, we provide the NMR signal in the Supplementary Material to show that the signal in the decay curves are strong and that the inverted water contents match those data.*


• Results: Line 265. Both the P and S wave interpretation show very little evidence of a "clear and observable feature" showing the recharge from the water table under the ephemeral stream. Would be much stronger if you report the Poisson's ratio in the results. It's an established technique – so doesn't need to go in the discussion. Much less emphasis on P and S Wave interpretation (unless to show that they are much inferior) and report the Poisson's
ratio – which is good.

*See general comments above. Good point. We will take these changes into account in the revised version of the manuscript.*

• Line 309 – you discuss no changes in lithology – however above you discuss clay below the water table – please clarify

*This is a good catch by the reviewer. We are convinced that the method worked well because there was no lithological*
*variation. That being said we also needed to convince ourselves and the readers that the higher Poisson's ratio was*
*due to an increase in saturation and not something else. An increase in clay content would give a similar response.*
*To fix this in the manuscript we will remove the reference to clay content at the end of the opening paragraph. In the*
*second sentence of the following paragraph we will clarify this statement with: "Although we don't expect any*
*lithological variation, it is possible that the region of high Poisson's ratios is a result of higher clay content since*
*materials that are deformed easily will have higher Poisson's ratios."*


    • Line 335 and following. This paragraph needs to be changed. You can't say "different physics" and "gravitate towards". Just say the second interpretation is more likely due to the NMR data.

*We agree. This was just trying to so that the second interpretation is more likely. This change will be implemented.*

    • Line 360 and following – first sentence you need to mention the observations from cores and piezometer. Also please revisit line 34 – Nano TEM identified low conductivity area – not an increase in saturation.

*This is a great suggestion and will be taken into account in the final revision of the manuscript.*

    • Line 375 an following. The use of Archie's equation here is questionable and weakens the paper. Youi have already mentioned very high and variable water conductivity and the presence of clays – both of which make applying Archie's equation unreliable. This detracts from the paper and I would delete this whole section

*Addressed in the general comment section above.*

    • Line 410 and following Hydrogeological interpretations. One question here that is not answered is whether this water helps sustain an aquatic ecosystem, or vegetation, or is it "lost" to a saline groundwater system.

*This is a great question and one that needs to remain open ended. We did not look at ET or any ecosystem related*
*data in this manuscript. One concern is that with the opening of a new water treatment plant that recycles treated*
*waste water for irrigation reuse in the horticultural industries, the application of new irrigation water could*
*eventually waterlog the soils and bring the much more salty groundwater to the surface. Here we might be able to*
*suggest that these small ephemeral features, visible only in LiDAR, are at higher risk.*


- line 441 and following, "impossible to know ahead of time"? Delete this. The geophys- ical survey particularly Poisson's ration and nanoTEM has helped confirm recharge occurring and therefore guide the siting of more detailed drilling.

*Agreed. We will delete this phrase.*

- Line 454 – New conceptual model of groundwater recharge in semi arid areas? Unless I missed something I don't think you have done this. Many have discussed recharge from ephemeral streams of all sizes – you have confirmed recharge has occurred from a very small "0" order tributary using a sensible and well applied combination of geo-physical methods and ground truthed with a piezometer.

*See the responses to the general comments. We agree and have addressed these concerns by removing language of a new conceptual model and instead extending the current model to smaller scale ephemeral recharge features.*

- Conclusions: please name check the Poisson's ratio – which proved useful Line 490 – Do they play a vital role in recharge to the NAP? I don't think you can conclude this, you can say that you have confirmed that ephemeral recharge occurs – but not how important or significant it is to the overall water balance.

*We agree and we have weakened the language in the conclusion section. Once again see our responses to the general comments section above.*

- Diagrams Overall could do with improving the quality. Figure 1 is really difficult to follow. I don't think we need all the panels. A location map (that's easy to follow – currently couldn't tell land from sea) than panel C which is the main information. A cross section may also be useful if you want to keep in the wider context Figure 2, 3 – combine and add in the poisson's ratio. Add in the location of the piezometer an ephemeral feature to all diagrams Figure 4 – need more information on how to interpret the residuals. I don't know whether its good or bad. Figure 5 – you need to explain the discrepancy between the geophysical data and the in situ data Figure 6 – delete the line for the water table – and add in the piezometer and point observation Figure 7 – Delete along with the section on Archie's law
  - *For Figure 1 we will remove panel d. We will change the underlying map to Hillshade all together since elevations aren't all that important; we were trying to show the ephemeral stream features. We will remove reference to the 47 data points, but have also added a reference to the report where the data exists in the text and will make it clear that we used the data to site the location of the survey.*
  - *We will combine Figures 2 and 3 and add Poisson's ratio. We will make sure that Poisson's ratio is the biggest of the three plots. We will remove the vertical gradient. It's that we like to look at P-wave data, but overall not important to the main message of the paper.*
  - *We will remove the residuals. They were low and the inversions were good—that was the point.*
  - *To the new Figure 5. We will add the piezometer point for the observation well and make the horizontal line much more transparent. We think it's valuable to have the line in the figure so that your eye can follow the flat water table and the Poisson's ratio contours—but I understand we do only know the water level at that point where the observation well is located along the survey line—hopefully these changes makes the figure more clear.*
  - *Figure 7 will be removed from the main text and moved to the Supplementary Material.*

Dear Dr. Inverarity,

Thank you for taking the time to read and assess our manuscript. Your major concern was with the Archie's law section in the discussion section. This was something that the first reviewer also commented on. We have taken in to consideration the first reviewer's comments and move this section on Archie's Law to the Supplementary Material but still wanted to provide an answer to your comments.

**Specific Comments**

1. How do you justify selecting single values for the Archie's Law exponents? There is a lot of published evidence that these parameters vary widely and the only reliable way to establish them is empirically for a given formation. Does using a realistic range of values from the published literature alter your Figure 7? You also use this to state it is "possible" for the electrical conductivity to be lower; what about using the potential ranges in these parameters to work out a probability that this is the case? It seems there are other likely explanations which should have more discussion, such as a change in lithology. You state at line 466 that this is not the case, i.e. no lateral lithological variation; why? Is that supported by the wider set of borehole data collected as part of this work? Is it not possible that the topographic feature indicates lateral variation in subsurface material?

*The objective of this section was to convince the readers (and ourselves) that it was possible to decrease the conductivity by increasing the saturation with a more resistive fluid. The section was never intended to be a fully statistical exploration; thus, we didn't explore a range of realistic values. Essentially, this section stemmed from the idea that we could actually see a rise in Poisson's Ratio due to a rise in clay content. This is stated in the second paragraph of Section 5.2, "it is possible that the region of high Poisson's ratios is a result of higher clay content since….". To make it clearer we would add that "although we don't' expect any lithologic variation" to the beginning of this sentence. We use the TEM data to rule out a change in lithology or specifically an increased clay content. If it were an increase in clay content we would have seen the ground get more conductive—but in reality we observed the opposite.*

*The Archie's Law modeling section was a thought experiment to show that it would be possible. It was never indented to be comprehensive. We also think that expanding this section would detract from the main objective of this section, which was to show that it is possible to see a decrease in conductivity. To address this comment in the manuscript we will move Figure 7 to the Supplementary Material and replace two paragraphs of our discussion on the topic with one sentence, "A simplified and general modelling exercise using Archie's Law shows that if we replace the water in the pores with a more resistive fluid, it is possible to get a drop in electrical conductivity even if the saturation is*

*increased (refer to Supplementary Material)." We will also remove the reference to the other 47 boreholes. In the Geologic Setting section, we will make it clear that we used these existing holes to select a site.*

2. I think the conclusion is too firm – without monitoring of the surface water and time- lapse data, I don't think you can demonstrate that it is a recharge feature, unless you can exclude the possibility that the Poisson's ratio and conductivity data are not both related to a lithological feature.

*We feel that we have demonstrated that the Poisson's ratio and conductivity data are not related to a lithologic feature. In summary we show that the P-wave and S-wave velocity data are high quality and that the rise in Poisson's ratio under the depression is real. In the second paragraph in Section 5.2, we specifically acknowledge that it could be caused by a change in lithology and use this section to show that the increase in Poisson's ratio is most likely caused by an increase in saturation. We use the partial saturation in the NMR profile, the decrease in conductivity, and the fact that the groundwater in the region is known to be highly saline. Thus, the assumption is that rainwater will be more resistive than groundwater—this seems reasonable.*

*We also acknowledge the limitations of our methods and that we can't definitely say there is recharge because we only have one measurement in time. In Section 5.3, "*It should be noted that our hydrogeological interpretation is based on a single snapshot in time. Without time-lapse geophysical measurements, groundwater samples taken from within the groundwater mound and either side, or long-term monitoring of groundwater observations wells, it is not possible to definitively quantify the recharge rates in these systems. Nor is it possible to determine if the groundwater mound is a result of a recent rainfall event or if it is a more stable feature.*" At best we have identified a water table bulge, which implies recharge, but we would need to see a response in the water table at the drillhole (which was not constructed as a monitoring well and now has been filled in) to confirm. We also acknowledge all these limitations in the second to last paragraph of Section 5.4. In other words we know that the presentation of data here is a unique tool and attempted to highlight limitations and assumptions for others if they wish to apply this combined geophysical technique elsewhere.*

**Technical Corrections**

- Line 31: "recharges" not "recharge". Lines 80, 107: "East" should be "east". Line 90: "sediments" not "rocks" (not lithified). Line 173: "S-wave" not "S-Wave". Line 340: remove "gravitate", use something like "prefer" instead? Line 393: "quaternary" should be "Quaternary"

*We will fix all of these minor corrections in the revised version of the manuscript.*