# Peer review of "Identifying recharge under subtle ephemeral features in flat-lying semi-arid region using a combined geophysical approach"

_Hydrology and Earth System Sciences, 2019_

## Short Comment (SC1) · 10 Jan 2020

We would love to invite discussion over our use of "capillary fringe". This language occurs on L298 and L351.

---

## Referee Comment (RC1) · Alan MacDonald (Referee) · 27 Feb 2020

General Comments

This is an interesting paper which applies several near surface geophysical techniques to show that recharge is occurring from a subtle ephemeral feature in Australia. Although arguably a case study, the combination of geophysical methods used and the narrow focus on a very small feature make it of interest to a wide audience researching dryland hydrology. I would recommend publication, however the paper would benefit from revisions to improve its clarity and impact.

[Figure]

General Comments (expanded in the Specific comments below): 1. Discuss and frame 
[revised manuscript text omitted]

---

## Referee Comment (RC2) · Kent Inverarity (Referee) · 19 Apr 2020

General comments

Thank you for the discussion paper. This is well-written with interesting data and presents a compelling case for geophysical imaging of the water table.

Specific comments

How do you justify selecting single values for the Archie's Law exponents? There is a lot of published evidence that these parameters vary widely and the only reliable way to establish them is empirically for a given formation. Does using a realistic range of

values from the published literature alter your Figure 7? You also use this to state it is "possible" for the electrical conductivity to be lower; what about using the potential ranges in these parameters to work out a probability that this is the case? It seems there are other likely explanations which should have more discussion, such as a change in lithology. You state at line 466 that this is not the case, i.e. no lateral lithological variation; why? Is that supported by the wider set of borehole data collected as part of this work? Is it not possible that the topographic feature indicates lateral variation in subsurface material?

I think the conclusion is too firm – without monitoring of the surface water and time-lapse data, I don't think you can demonstrate that it is a recharge feature, unless you can exclude the possibility that the Poisson's ratio and conductivity data are not both related to a lithological feature.

Technical corrections

Line 31: "recharges" not "recharge". Lines 80, 107: "East" should be "east". Line 90: "sediments" not "rocks" (not lithified). Line 173: "S-wave" not "S-Wave". Line 340: remove "gravitate", use something like "prefer" instead? Line 393: "quaternary" should be "Quaternary"

---

## Author Comment (AC2) · 8 May 2020

Dear Dr. Inverarity,

Thank you for taking the time to read and assess our manuscript. Your major concern was with the Archie's law section in the discussion section. This was something that the first reviewer also commented on. We have taken in to consideration the first reviewer's comments and move this section on Archie's Law to the Supplementary Material but still wanted to provide an answer to your comments.

Specific Comments

1. How do you justify selecting single values for the Archie's Law exponents? There is a lot of published evidence that these parameters vary widely and the only reliable way to establish them is empirically for a given formation. Does using a realistic range of values from the published literature alter your Figure 7? You also use this to state it is "possible" for the electrical conductivity to be lower; what about using the potential ranges in these parameters to work out a probability that this is the case? It seems there are other likely explanations which should have more discussion, such as a change in lithology. You state at line 466 that this is not the case, i.e. no lateral lithological variation; why? Is that supported by the wider set of borehole data collected as part of this work? Is it not possible that the topographic feature indicates lateral variation in subsurface material?

The objective of this section was to convince the readers (and ourselves) that it was possible to decrease the conductivity by increasing the saturation with a more resistive fluid. The section was never intended to be a fully statistical exploration; thus, we didn't explore a range of realistic values. Essentially, this section stemmed from the idea that we could actually see a rise in Poisson's Ratio due to a rise in clay content. This is stated in the second paragraph of Section 5.2, "it is possible that the region of high Poisson's ratios is a result of higher clay content since....". To make it clearer we would add that "although we don't' expect any lithologic variation" to the beginning of this sentence. We use the TEM data to rule out a change in lithology or specifically an increased clay content. If it were an increase in clay content we would have seen the ground get more conductive—but in reality we observed the opposite.

The Archie's Law modeling section was a thought experiment to show that it would be possible. It was never indented to be comprehensive. We also think that expanding this section would detract from the main objective of this section, which was to show that it is possible to see a decrease in conductivity. To address this comment in the manuscript we will move Figure 7 to the Supplementary Material and replace two paragraphs of our discussion on the topic with one sentence, "A simplified and general modelling

exercise using Archie's Law shows that if we replace the water in the pores with a more resistive fluid, it is possible to get a drop in electrical conductivity even if the saturation is increased (refer to Supplementary Material)." We will also remove the reference to the other 47 boreholes. In the Geologic Setting section, we will make it clear that we used these existing holes to select a site.

2. I think the conclusion is too firm – without monitoring of the surface water and time-lapse data, I don't think you can demonstrate that it is a recharge feature, unless you can exclude the possibility that the Poisson's ratio and conductivity data are not both related to a lithological feature.

We feel that we have demonstrated that the Poisson's ratio and conductivity data are not related to a lithologic feature. In summary we show that the P-wave and S-wave velocity data are high quality and that the rise in Poisson's ratio under the depression is real. In the second paragraph in Section 5.2, we specifically acknowledge that it could be caused by a change in lithology and use this section to show that the increase in Poisson's ratio is most likely caused by an increase in saturation. We use the partial saturation in the NMR profile, the decrease in conductivity, and the fact that the ground-water in the region is known to be highly saline. Thus, the assumption is that rainwater will be more resistive than groundwater—this seems reasonable.

We also acknowledge the limitations of our methods and that we can't definitely say there is recharge because we only have one measurement in time. In Section 5.3, "It should be noted that our hydrogeological interpretation is based on a single snapshot in time. Without time-lapse geophysical measurements, groundwater samples taken from within the groundwater mound and either side, or long-term monitoring of groundwater observations wells, it is not possible to definitively quantify the recharge rates in these systems. Nor is it possible to determine if the groundwater mound is a result of a recent rainfall event or if it is a more stable feature." At best we have identified a water table bulge, which implies recharge, but we would need to see a response in the water table at the drillhole (which was not constructed as a monitoring well and now has been

filled in) to confirm. We also acknowledge all these limitations in the second to last paragraph of Section 5.4. In other words we know that the presentation of data here is a unique tool and attempted to highlight limitations and assumptions for others if they wish to apply this combined geophysical technique elsewhere.

Technical Corrections - Line 31: "recharges" not "recharge". Lines 80, 107: "East" should be "east". Line 90: "sediments" not "rocks" (not lithified). Line 173: "S-wave" not "S-Wave". Line 340: remove "gravitate", use something like "prefer" instead? Line 393: "quaternary" should be "Quaternary"

We will fix all of these minor corrections in the revised version of the manuscript.

---

## Author Response (AR1)

[revised manuscript text omitted]

Field Code Changed

Field Code Changed

Commented [EB5]: Just a general comment but isn't the water quality more of a dominant factor in the TEM response than a slight change in lithologyl

[revised manuscript text omitted]